# Effects of Zn Content on Hot Tearing Susceptibility of Mg–7Gd–5Y–0.5Zr Alloy

**Ziqi Wei** [1,2] , **Shimeng Liu** [1,2,*], **Zheng Liu** [1,2], **Feng Wang** [1,2], **Pingli Mao** [1,2], **Xiaoxia Wang** [1,2] **and Xingxing Li** [1,2]

[1] School of Materials Science and Engineering, Shenyang University of Technology, Shenyang 110870, China; weiziq@126.com (Z.W.); zliu4321@126.com (Z.L.); wf9709@126.com (F.W.); pinglimao@yahoo.com (P.M.); xiaoxiawang95@126.com (X.W.); xxli4321@126.com (X.L.)

[2] Key Laboratory of Magnesium Alloys and the Processing technology of Liaoning Province, Shenyang 110870, China

[*] Correspondence: liushimeng123456@163.com; Tel.: +86-024-25466301

**Abstract:** Mg–7Gd–5Y–0.5Zr alloy has excellent mechanical properties but poor hot tearing resistance. The latter makes it difficult to cast billets, which limits the size of subsequently processed parts. Therefore, the hot tearing susceptibility of Mg–7Gd–5Y–$x$Zn–0.5Zr ($x$ = 0, 3, 5, 7 wt%) alloys was studied. It was found that Zn can significantly reduce hot tearing susceptibility of Mg–7Gd–5Y–0.5Zr alloy, which almost linearly decreased with Zn content. When Zn content was 3 wt%, 5 wt% and 7 wt%, hot tearing susceptibility will be reduced by 27%, 83% and 100%, respectively. It was further revealed that the solid solubility of Gd and Y in $\alpha$-Mg decreased with the increase of Zn content, and the nucleation temperature decreased accordingly, which resulted in the increase of nucleation rate and the refinement of final grains. On the macro level, it showed that the dendrite coherency temperature decreased, the solidification shrinkage stress of $\alpha$-Mg slowed down, and the residual liquid channel became shorter and hot tearing susceptibility decreased. It was also found that with the increase of Zn content, the content of Zn, Gd and Y enriched on the grain boundary increased, the content of residual liquid phase between dendrites increased after $\alpha$-Mg crystallization, and the solidified precipitated second phase also changed from $Mg_5RE$ phase to long-period stacking ordered phase + W-phase (a little), long-period stacking ordered phase + W-phase (much) and finally to W-phase only. The feeding effect of sufficient intergranular residual liquid on the shrinkage of $\alpha$-Mg dendrite and the bridging effect of the precipitated phase at the grain boundary (especially long-period stacking ordered phase which is coherent with the parent phase) also led to the decrease of hot tearing susceptibility.

**Keywords:** hot tearing susceptibility; Mg–Gd–Y–Zn–Zr alloy; microstructure; precipitation phase

## 1. Introduction

Hot tearing is often a major casting defect in magnesium alloys and has a significant impact on the quality of their casting products. The research on hot tearing of magnesium alloy has made a lot of progress in recent decades [1], and the further research is still needed to understand more about the behavior and mechanism of hot tearing. Especially for some newly developed high-performance magnesium alloys, it is equally important to understand hot tearing and mechanical properties.

Mg–Gd–Y–Zr alloys have the advantages of high strength and good creep resistance due to the high content of rare earth elements [2–5]. Therefore, the alloys have very wide application prospects in aerospace and other special fields [6–8]. However, one of the unavoidable shortcomings of the alloys

with high rare earth content is the high hot tearing tendency during solidification, which greatly limits the size of their ingots and the structural complexity of casting parts [9–12].

In recent years, it has been found that adding Zn to Mg–Gd–Y–Zr alloys can further improve its mechanical properties by the precipitation of long-period stacking ordered (LPSO) phases [13–17]. Chi et al. [18] found that tensile yield strength (TYS) of the extruded specimens increased from 287 to 330 MPa, and ultimate tensile strength (UTS) increased from 342 to 392 MPa with the addition of 0.5 wt% Zn into the Mg–8.5Gd–2.5Y–0.3Zr alloy. Wang et al. [19] further studied the effect of the Zn addition on the mechanical properties of Mg–10.5Gd–5Y–0.5Zr alloy. When the Zn content increased from 0.65 to 1.2 wt%, the TYS of the alloy increased from 284 to 286 MPa, and the UTS increased from 421 to 424 MPa. When the Zn content further increased to 1.8 wt%, the TYS of the alloy increased to 297 MPa, while the UTS of the alloy did not continue to increase but decreased to 405 MPa. However, it is not clear how the addition of Zn and its content effect hot tearing susceptibility (HTS) of Mg–Gd–Y–Zr alloys.

Mg–7Gd–5Y–0.5Zr alloy is a new type of magnesium alloy with high strength, high toughness and high thermal stability developed in recent years [20–23]. However, it shows a high hot tearing tendency during ingot casting. Gunde et al. [24] found that Y can reduce the HTS of Mg–Zn alloy by shortening the solidification path, whereas the effect of Zn on the hot tearing of Mg–RE alloy and the micro mechanism are not clear. For this reason, the authors are inspired by the results that Zn can improve the mechanical properties of Mg–Gd–Y–Zr alloys and try to study the effect of Zn content on HTS of Mg–7Gd–5Y–0.5Zr alloy. In the paper, based on the Clyne-Davies' model and the solidification thermal analysis method, the HTS of Mg–7Gd–5Y–0.5Zr magnesium alloy with different Zn content (0, 3, 5, 7 wt%) was predicted. The predicted results were further validated by the constrained rod casting (CRC) method with time, temperature and load signal acquisition system. It has been proved that, just as Bichler and Ravindran [25] thought, it is impossible to identify the hot tearing propagation by any hot tearing criterion. Therefore, the micro-mechanism of hot tearing formation was discussed based on the measured solidification characteristic parameters, as well as the observation of the microstructure and crack morphology of the alloys.

## 2. Experimental Procedures

The experimental Mg–7Gd–5Y–$x$Zn–0.5Zr ($x$ = 0, 3, 5, 7 wt%) alloys were prepared using a boron nitride (BN) coated crucible furnace by melting pure Mg, pure Zn, Mg–25Y (wt%), Mg–30Gd (wt%) and Mg–30Zr (wt%) in the protective 99.8% $N_2$/0.2% $SF_6$ mixed gas environment. The alloys were prepared by permanent "T"-shaped mold casting. The alloy melts were stirred for 3 min and then held at 720 °C for 30 min to ensure dissolution and homogenization of the alloy elements. Then the molten metal was poured into the T-shaped mold preheated of 250 °C. The actual compositions of Mg–7Gd–5Y–$x$Zn–0.5Zr alloys are listed in Table 1.

**Table 1.** The actual compositions of as-cast Mg–7Gd–5Y–$x$Zn–0.5Zr alloys (wt%).

| Alloy No. | Alloy Composition | Mg | Gd | Y | Zn | Zr |
|---|---|---|---|---|---|---|
| Alloy I | Mg–7Gd–5Y–0.5Zr | Bal. | 7.12 | 4.98 | 0 | 0.45 |
| Alloy II | Mg–7Gd–5Y–3Zn–0.5Zr | Bal. | 7.10 | 4.82 | 3.21 | 0.48 |
| Alloy III | Mg–7Gd–5Y–5Zn–0.5Zr | Bal. | 7.06 | 4.94 | 4.89 | 0.44 |
| Alloy IV | Mg–7Gd–5Y–7Zn–0.5Zr | Bal. | 7.18 | 5.29 | 6.94 | 0.46 |

Figure 1 shows a T-shaped hot tearing acquisition device used to capture the shrinkage force and temperature of hot tearing initiation and propagation of the experimental alloys. When the molten alloys were poured into the T-shaped mold and cooled in air, the intersection between the sprue and the connecting rod was prone to stress concentration [26,27]. Generally, hot tearing may occur both inside and outside, and it is more difficult to evaluate internal tears than external tears. Therefore, the test mold is specially designed as a T-shape, so that only the external crack occurs at the corner of

the T-shaped mold with the most concentrated thermal stress, and then HTS of the alloys with different compositions could be easily compared and evaluated.

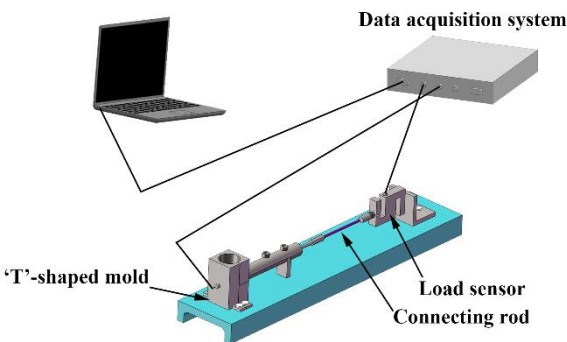

**Figure 1.** T-shaped hot tearing acquisition device.

Figure 2 shows the schematic diagram of differential thermal analysis used to characterize the solidification path of the experimental alloys. As solidification proceeds, two thermocouples were used to measure the center ($T_c$) and edge ($T_e$) temperature of the melt, respectively. Based on the cooling curve and its first derivative, the Newton baseline method can be used to calculate liquid fraction at different solidification temperatures, and some characteristic values such as the precipitation temperature of the second phase and the amount of precipitation of alloy solidification can also be obtained [28,29]. In addition, the temperature of α-Mg dendrite coherency ($T^{\alpha}_{coh}$) and solidification fraction of α-Mg dendrite coherency ($f^{\alpha}_{coh}$) can be measured to confirm HTS of alloys by the difference between the temperature measured by two thermocouples [30,31]. There was a difference ($\Delta T$) between $T_e$ and $T_c$ due to different solidification speed of melt center and edge. When the α-Mg dendrite coherency, the heat conduction in solid phase was much faster than that in liquid phase, $T_e$ and $T_c$ will approach rapidly, that is, a minimum value of $\Delta T$ will appear, and the $T_c$ was defined as $T^{\alpha}_{coh}$.

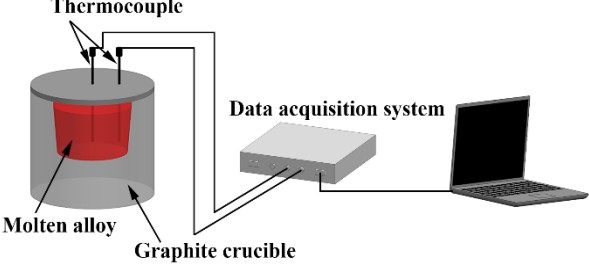

**Figure 2.** Schematic diagram of differential thermal analysis.

According to Clyne-Davies [32,33], as the liquid phase enclosed in the local space between dendrites leads to the solidification shrinkage volume being difficult to be fed, the liquid fraction between 0.01 and 0.1 is called the vulnerable time interval ($t_V$); as the liquid phase is flowing freely, the solidification shrinkage volume can be fed at any time, the liquid fraction between 0.1 and 0.6 is called the stress relaxation time interval ($t_R$). The ratio of $t_V$ and $t_R$, i.e., cracking susceptibility coefficient (CSC) value, can be calculated to predict HTS of the alloys. The mathematical expression of CSC is as follows:

$$\text{CSC} = \frac{t_V}{t_R} = \frac{t_{0.01} - t_{0.1}}{t_{0.1} - t_{0.6}} \tag{1}$$

where $t_V$ is the solidification time when the liquid fraction is between 0.01 and 0.1; $t_R$ is the solidification time when the liquid fraction is between 0.1 and 0.6.

Samples for microstructure observation were taken from the regions near hot tearing region or T-shaped hot junction. The phase composition of the investigated alloys was analyzed by an X-ray diffraction (XRD, D/Max-IIIA-Rigaku, Tokyo, Japan) with a copper target, in scanning angle range of 20° to 90°, at a scanning speed of 2°/min. The microstructures of the samples were determined by using scanning electron microscope (SEM, S-3400N-Hitachi, Tokyo, Japan), electron back scatter diffraction (EBSD, GeminiSEM 300-Zeiss, Jena, Germany) and transmission electron microscope (TEM, JEM-2100-JEOL, Tokyo, Japan). As-cast samples for SEM observation were ground, polished and etched in 4% $HNO_3$ with ethanol. After mechanical polishing, EBSD samples were electrolytically-polished in an electrolyte of 10% perchloric acid and 90% ethanol at 15 V and −30 °C for 90 s. EBSD data was analyzed by using Channel 5 software (HKL, Oxford, UK). TEM samples were first ground to 80 μm, then punched into discs with 3 mm in diameter and ground to 50 μm followed by low angle ion milling using Gatan precision ion polishing system.

## 3. Results and Discussion

### 3.1. Phase and Microstructure in As-Cast Condition

#### 3.1.1. Phase Composition and Microstructure

It has been found that in Mg–Zn–RE alloy, the type of the second phase precipitated during solidification was determined by the ratio of Zn/RE [34], which also effected HTS of the alloy [35]. Figure 3 shows XRD results of Mg–7Gd–5Y–$x$Zn–0.5Zr alloys. It was found that there was overlap between LPSO phase and parent phase, and the diffraction peaks of LPSO phase and W-phase moved to high angle direction with the increase of Zn content. According to the diffraction peaks, the phase composition of the four alloys was about $\alpha$-Mg and one or two second phases. When Zn was not contained, the second phase in the alloy can be determined as only $Mg_5RE$. When Zn content was 3 or 5 wt%, the second phase should be composed of LPSO phase ($Mg_{12}Zn(Gd,Y)$) and W-phase ($Mg_3Zn_3(Gd,Y)_2$). When Zn content increased to 7 wt%, LPSO phase disappeared completely, and the second phase should be only W-phase.

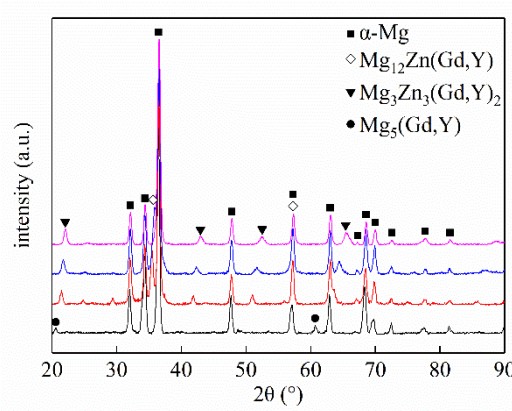

**Figure 3.** XRD results of Mg–7Gd–5Y–$x$Zn–0.5Zr alloys.

Some diffraction peaks of the parent phase and the second phase coincide, therefore the X-ray diffraction results need to be verified by other methods. Figure 4 shows microstructures and morphology of Mg–7Gd–5Y–$x$Zn–0.5Zr alloys. It can be seen that the phase compositions of the four alloys were grey block $\alpha$-Mg and one or two second phases precipitated along the grain boundary of the parent phase. As shown in Figure 4a, the second phase was $Mg_5(Gd,Y)$, which was a bright white fishbone phase and distributed at $\alpha$-Mg grain boundary, as shown by the arrow. As shown in Figure 4b,c, the second phases were LPSO phase and W-phase, which were distributed in the grain boundary. As shown in Figure 4d, the second phase was almost all W-phase, which was distributed in the grain

boundary in the form of network. The volume fraction of LPSO phase and W-phase in the second phase under different Zn content in Figure 4 was calculated by Image Pro Plus software. It was found that when Zn content was 3 wt%, the corresponding LPSO phase fraction was 18.3%, W-phase fraction was 4.6%; when Zn content was 5 wt%, the corresponding LPSO phase fraction was 21.2%, W-phase fraction was 12.6%; when Zn content was 7 wt%, the corresponding LPSO phase almost disappeared, W-phase fraction was 35.9%. It can also be seen from Figure 4 that the final grains size decreased significantly with the increase of Zn content, which should be beneficial to the reduction of HTS [10,36].

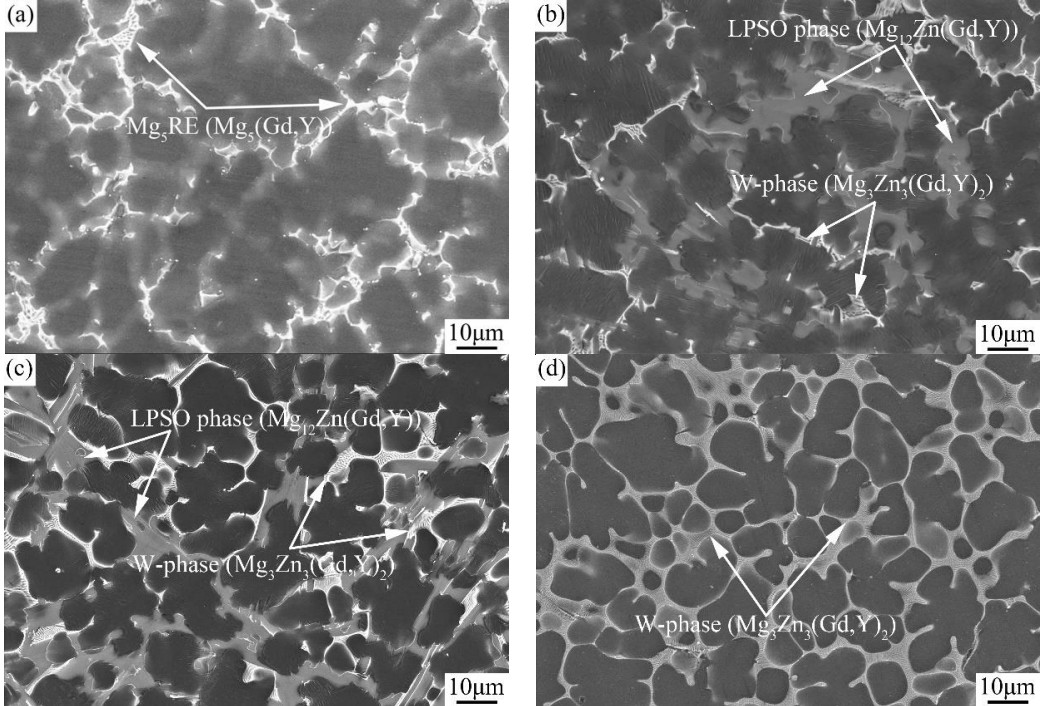

**Figure 4.** Microstructure and morphology of Mg–7Gd–5Y–*x*Zn–0.5Zr alloys: (**a**) *x* = 0 wt%; (**b**) *x* = 3 wt%; (**c**) *x* = 5 wt%; (**d**) *x* = 7 wt%.

### 3.1.2. Thermal Analysis Results

The solidification path of the above-mentioned final structure can be clearly displayed by thermal analysis method, so that the change of liquid fraction with time can be further calculated by Newton baseline method on this basis, and the required data can be provided for the prediction of HTS by Clyne-Davies' criterion. Figure 5 shows the thermal analysis results of Mg–7Gd–5Y–*x*Zn–0.5Zr alloys during solidification. The peaks on the first derivative curve based on the cooling curve were caused by the latent heat released from the phase transformation process. The area between baseline and exothermic peak was proportional to the heat released by the phase precipitation and hence to the relative precipitated amount [37–39]. The characteristic temperatures on the solidification path of the alloys are summarized in Table 2.

**Table 2.** The temperatures of phase transformation of Mg–7Gd–5Y–*x*Zn–0.5Zr alloys solidification path in Figure 5 (temperature, °C).

| Alloy | L→α-Mg | L→α-Mg + Mg₅RE | L→α-Mg + LPSO | L→α-Mg + W | $T_s$ |
|---|---|---|---|---|---|
| Alloy I | 630 | 547 | —— | —— | 537 |
| Alloy II | 623 | —— | 523 | 517 | 507 |
| Alloy III | 619 | —— | 516 | 510 | 492 |
| Alloy IV | 619 | —— | —— | 519 | 509 |

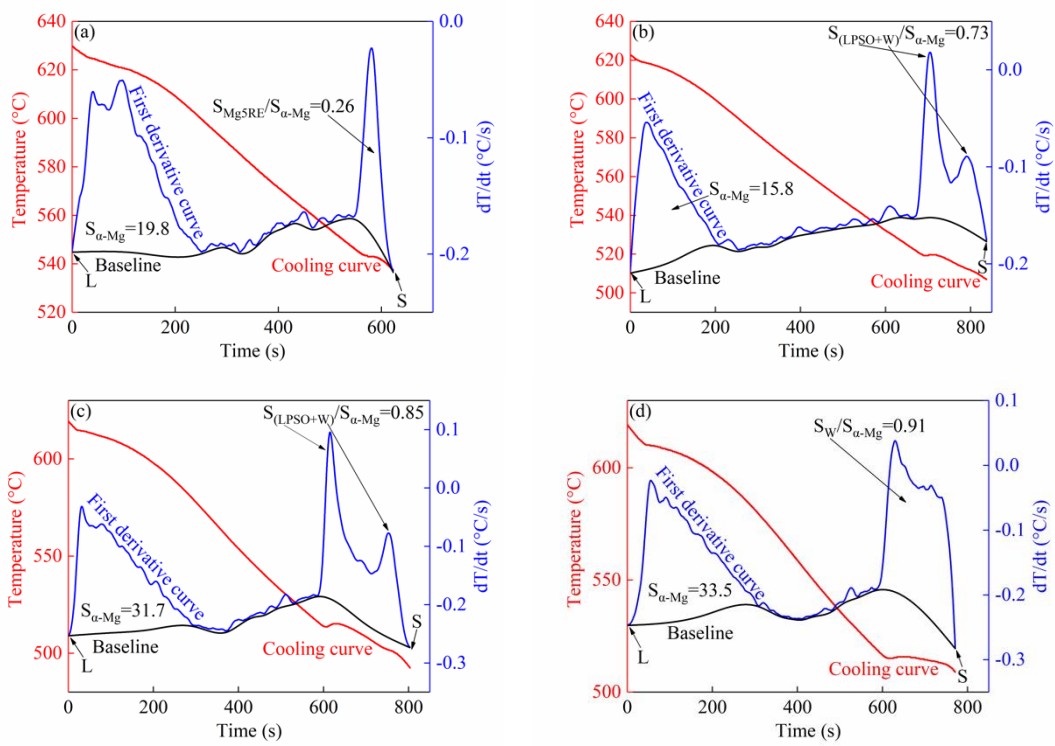

**Figure 5.** Solidification path of Mg–7Gd–5Y–*x*Zn–0.5Zr alloys: (**a**) *x* = 0 wt%; (**b**) *x* = 3 wt%; (**c**) *x* = 5 wt%; (**d**) *x* = 7 wt%.

It can be seen from Figure 5a that there were two exothermic peaks in the first derivative curve of alloy I which was $\alpha$-Mg, $Mg_5RE$ phase, respectively, and the area ratio of $Mg_5RE$ phase to $\alpha$-Mg was 0.26. It can be seen from Figure 5b that there were three exothermic peaks in the first derivative curve of alloy II, which were $\alpha$-Mg, LPSO phase and W-phase. The area ratio of LPSO phase and W-phase relative to $\alpha$-Mg was 0.73. It can be seen from Figure 5c that there were three exothermic peaks in the first derivative curve of alloy III, which were $\alpha$-Mg, LPSO phase and W-phase. The area ratio of LPSO phase and W-phase to $\alpha$-Mg increased to 0.85. In Figure 5d, there were two exothermic peaks in the first derivative curve of alloy IV, which meant that the second phase precipitated was only W-phase. The area ratio of W-phase to $\alpha$-Mg reached 0.91, indicating that the maximum amount of W-phase. By the way, it can also be seen from Table 2 that the initial nucleation temperature of $\alpha$-Mg decreased with the increase of Zn content, which may be caused by the increase of the equilibrium concentration difference between the solid and liquid phases caused by select crystallization, and the higher concentration fluctuation required for solid phase nucleation. This may be one of the reasons for the final grains refinement of the alloy with the increase of Zn content as shown in Figure 4.

### 3.2. Solidification Characteristics and Hot Tearing Susceptibility

#### 3.2.1. Solidification Characteristics

There was no doubt that the effect of the increase of Zn content on the crystallization mode of the parent phase will also be reflected in its solidification path. According to the temperature ($T^\alpha_{coh}$) or solid fraction ($f^\alpha_{coh}$) corresponding to the dendrite coherent time, the solidification of alloy can be divided into two stages, which can be used to characterize the $\alpha$-Mg behavior [30]. Generally, the higher the $T^\alpha_{coh}$ (or the lower the $f^\alpha_{coh}$) is, the longer the stage of intergranular feeding relative to free feeding is. Figure 6 shows the change of $T^\alpha_{coh}$ and $f^\alpha_{coh}$ with Zn content in Mg–7Gd–5Y–*x*Zn–0.5Zr alloys. It can be seen that when Zn content was 0 wt%, 3 wt%, 5 wt% and 7 wt%, $T^\alpha_{coh}$ was 617, 610, 597 and 588 °C, which corresponded to $f^\alpha_{coh}$ of 72.9%, 82.3%, 90.1% and 92.5%, respectively.

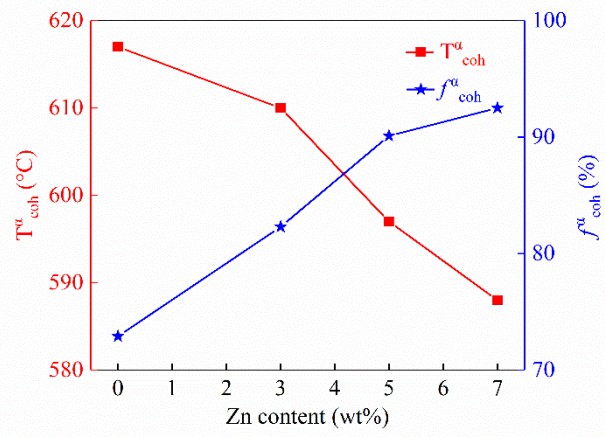

**Figure 6.** The change of $T^{\alpha}_{coh}$ and $f^{\alpha}_{coh}$ with Zn content.

Furthermore, the EBSD pole figures of Mg–7Gd–5Y–*x*Zn–0.5Zr alloys are shown in Figure 7. Obviously, the four alloys all had the crystallographic texture which was generally considered as {0001} basal plane was parallel to the direction of heat flow [36], but the texture strength decreased with the increase of Zn content. This showed that, on the one hand, the non-close-packed surface perpendicular to the close-packed surface had a preferential growth trend, but it decreased with the increase of the solute atom segregation at the solidification front. On the other hand, with the expansion of the subcooled region of the composition, the driving force for the reverse heat flow growth of the dendrites was reduced, and the front equiaxed crystal nucleation rate was increased. It can be inferred that the change of $\alpha$-Mg crystallization mechanism with the increase of solute interface segregation may be an important reason for the decrease of $T^{\alpha}_{coh}$ (or the increase of $f^{\alpha}_{coh}$) as shown in Figure 6 and the refinement of final grains as shown in Figure 4.

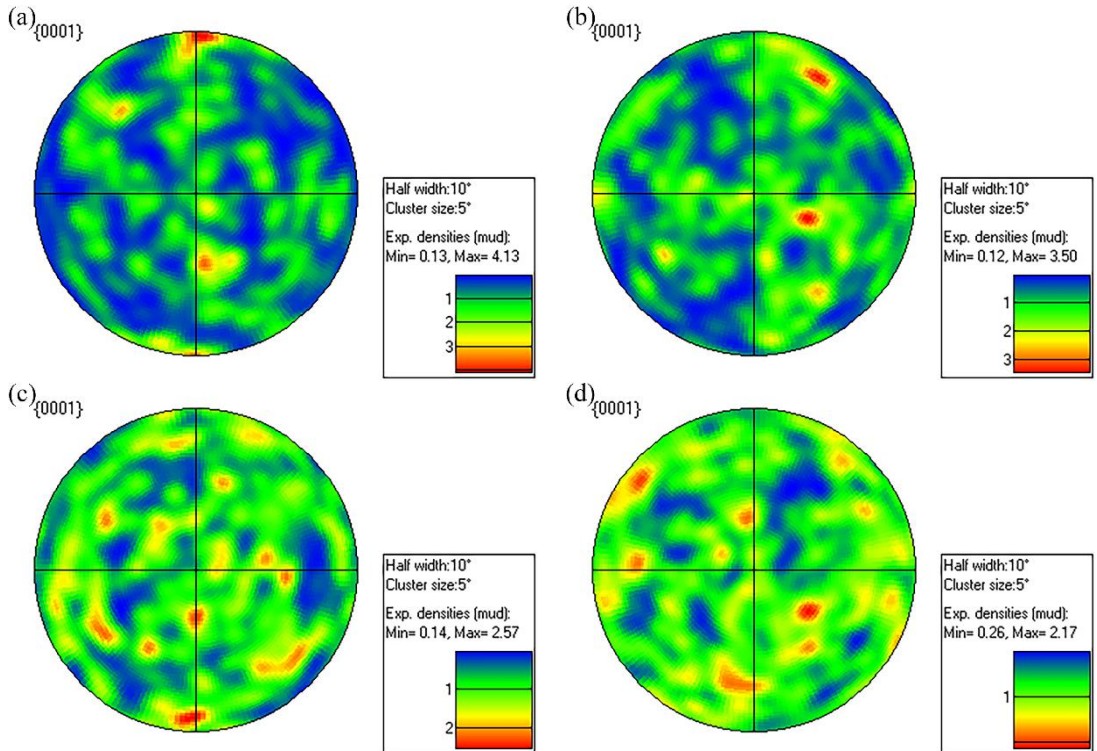

**Figure 7.** Electron back scatter diffraction (EBSD) pole figures of Mg–7Gd–5Y–*x*Zn–0.5Zr alloys: (**a**) *x* = 0 wt%; (**b**) *x* = 3 wt%; (**c**) *x* = 5 wt%; (**d**) *x* = 7 wt%.

### 3.2.2. Hot Tearing Behavior

Figure 8 shows the relationship between shrinkage force (*F*) and time (*t*) as well as temperature and time during solidification of T-shaped alloy specimens. Once the stress relaxation occurred on the F-t curve, it meant that cracks begin to initiate on the T-shaped specimen. A decline appeared on the shrinkage force curves when stress relaxation generated, which implied hot tearing initiated at the intersection of the T-shaped mold [40,41]. The characteristic parameters of hot tearing behavior extracted from Figure 8 are shown in Table 3. In addition, in order to eliminate the chance of experiments, each group of experiments was repeated at least three times.

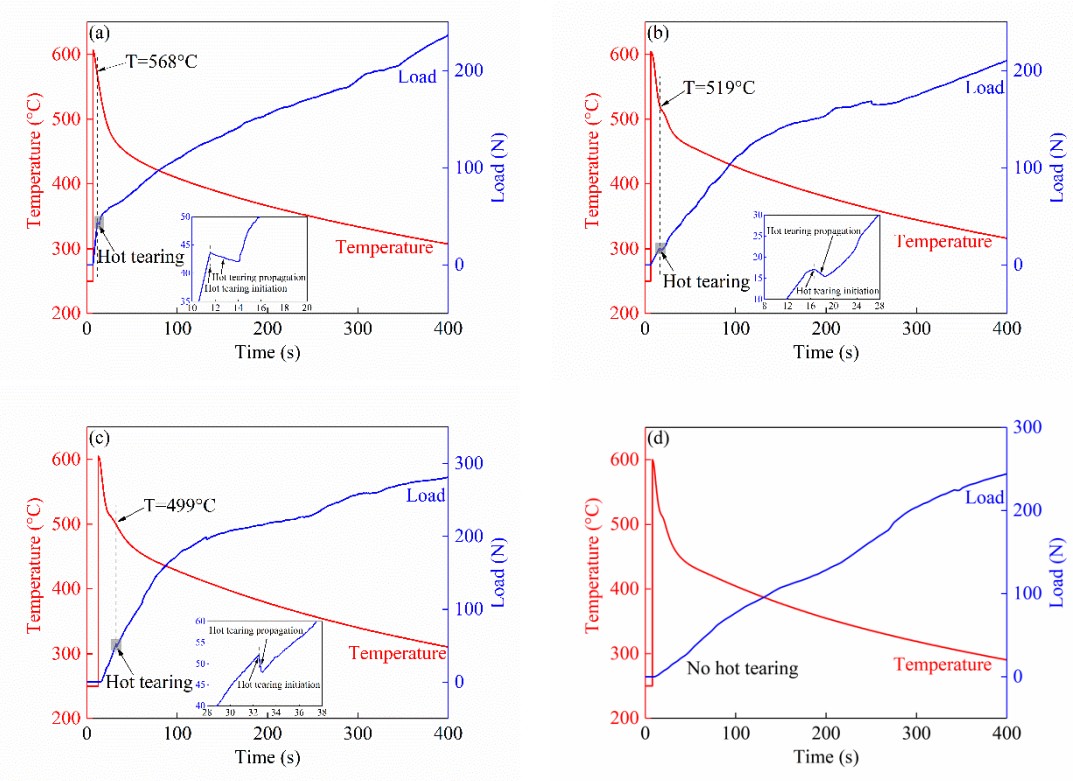

**Figure 8.** Hot tearing curves of Mg–7Gd–5Y–*x*Zn–0.5Zr alloys: (**a**) *x* = 0 wt%; (**b**) *x* = 3 wt%; (**c**) *x* = 5 wt%; (**d**) *x* = 7 wt%.

**Table 3.** Test value/standard deviation of hot tearing parameters of Mg–7Gd–5Y–*x*Zn–0.5Zr alloys.

| Alloy | Hot Tearing Initiation Time (s) | Hot Tearing Propagation End Time (s) | Hot Tearing Propagation Time Range (s) | Hot Tearing Initiation Temperature (°C) | Hot Tearing Propagation End Temperature (°C) |
|---|---|---|---|---|---|
| Alloy I | 11.6/0.65 | 14.1/0.17 | 2.5/0.62 | 568/3.86 | 546/0.94 |
| Alloy II | 16.6/0.08 | 18.4/0.29 | 1.8/0.24 | 519/0.94 | 514/2.16 |
| Alloy III | 32.5/0.29 | 32.8/0.22 | 0.3/0.17 | 499/3.10 | 498/2.49 |

As shown in Figure 8a, the hot tearing of alloy I occurred in the period of 568–546 °C (close to the precipitation temperature 547 °C of $Mg_5RE$ phase), and $Mg_5RE$ phase almost did not precipitate, indicating that there should be some residual liquid phase on the fracture surface during hot tearing. It can be seen from Figure 8b that hot tearing initiation of alloy II occurred at 519 °C, which was lower than the precipitation temperature of LPSO phase and slightly higher than that of W-phase. This meant that the LPSO phase precipitated before hot tearing, which will play a bridging role on the grain boundary and delay the occurrence of hot tearing. As shown in Figure 8c, the initial hot tearing temperature occurred at 499 °C, which was not only lower than the precipitation temperature

of LPSO phase, but also lower than that of W-phase. This meant that the effect of precipitation on grain boundary bridging was further enhanced. As shown in Figure 8d, no hot tearing occurred until the end of solidification for alloy IV.

### 3.2.3. Hot Tearing Susceptibility

Figure 9 shows the theoretically CSC values calculated based on the Clyne-Davies' model for predicting HTS of Mg–7Gd–5Y–$x$Zn–0.5Zr alloys. The various parameters such as $t_{0.01}$ were determined by thermal analysis method, and then the parameters were brought into Equation (1) to calculate the CSC values. It can be seen, CSC values of the investigated Mg–7Gd–5Y–$x$Zn–0.5Zr ($x$ = 0, 3, 5, 7 wt%) alloys were 0.88, 0.64, 0.15 and 0, respectively. The HTS of the tested alloys predicted by CSC values were obtained in the following order: Mg–7Gd–5Y–0.5Zr > Mg–7Gd–5Y–3Zn–0.5Zr > Mg–7Gd–5Y–5Zn–0.5Zr > Mg–7Gd–5Y–7Zn–0.5Zr. It can be seen that when Zn content was 3 wt%, 5 wt% and 7 wt%, HTS can be reduced by 27%, 83% and 100%, respectively, that is to say, Zn can significantly reduce HTS of Mg–7Gd–5Y–0.5Zr alloy, which almost linearly decreased with the Zn content.

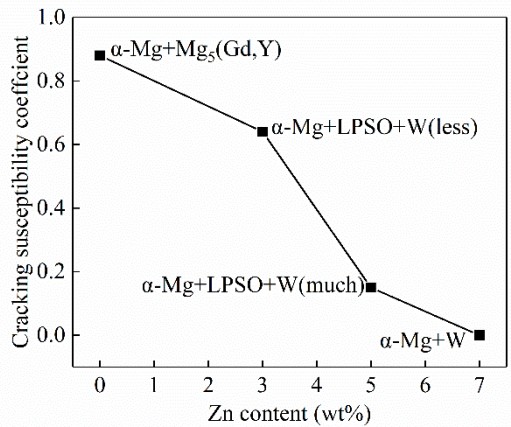

**Figure 9.** Predicted CSC values of Mg–7Gd–5Y–$x$Zn–0.5Zr alloys.

When the alloy solidifies in the T-shaped mold, hot tearing was most likely to occur at the corner of the mold where the thermal stress was the most concentrated [26,27]. Figure 10 shows the macro-tears at the T-shaped hot junction of Mg–7Gd–5Y–$x$Zn–0.5Zr alloy castings. The hot tearing volume of the alloys was measured by paraffin permeation method [42]. The values were 0.18, 0.11, 0.02 and 0 cm$^3$. It indicated that the HTS of the Mg–7Gd–5Y–$x$Zn–0.5Zr alloys decreased as the Zn content increased. It can be seen that, compared with the prediction value of CSC, they were in good agreement.

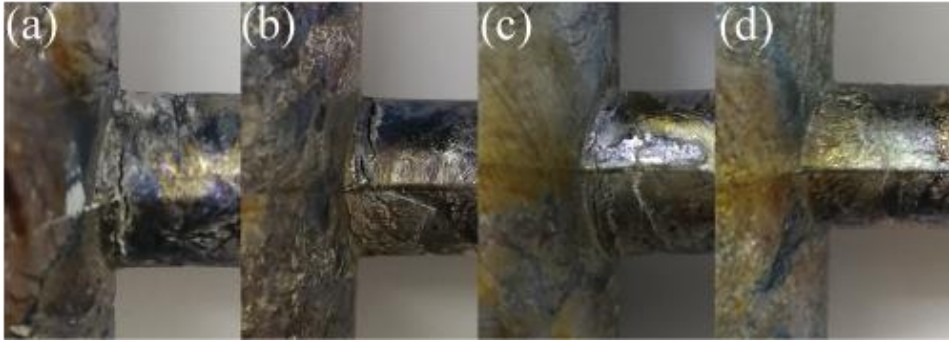

**Figure 10.** Macroscopic external hot tears of Mg–7Gd–5Y–$x$Zn–0.5Zr alloys: (**a**) $x$ = 0 wt%; (**b**) $x$ = 3 wt%; (**c**) $x$ = 5 wt%; (**d**) $x$ = 7 wt%.

### 3.3. Fracture Characteristics and Hot Tearing Mechanism

### 3.3.1. Fracture Characteristics

Figure 11 shows the micrographs near to the hot tearing region of T-shaped specimen of Mg–7Gd–5Y–*x*Zn–0.5Zr alloys. It can be seen from Figure 11a–c that the cracks began from the surface of the specimen to penetrate into the interior, and the crack size decreased with the increase of Zn content.

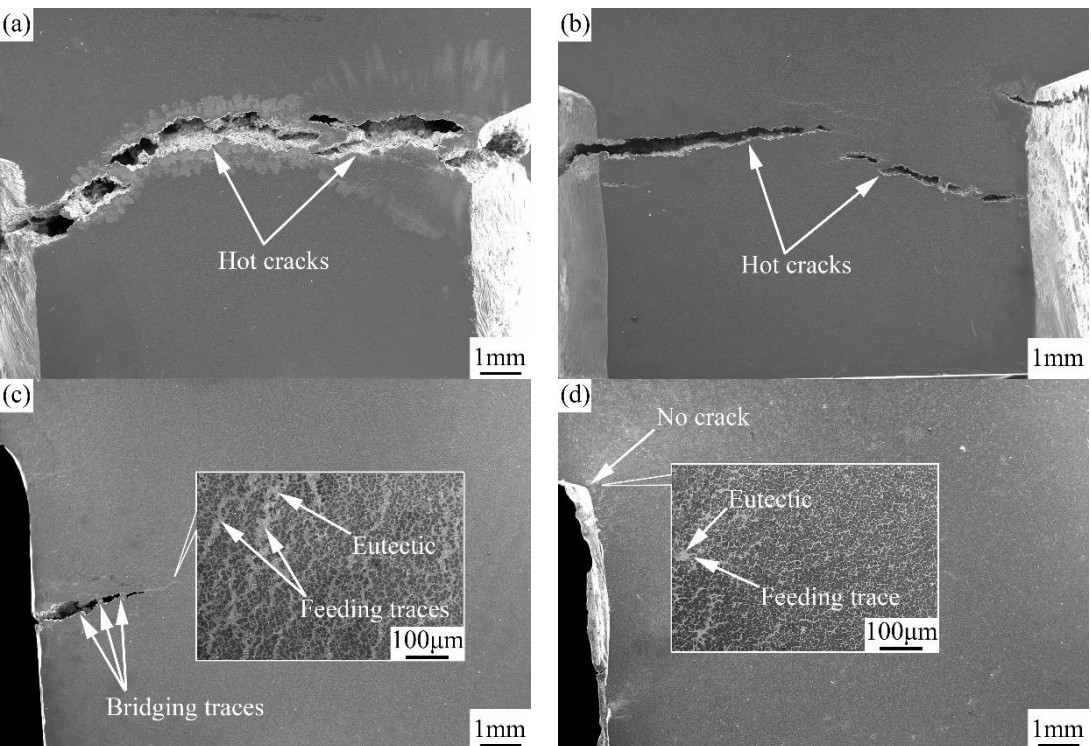

**Figure 11.** Micrographs of hot tears near T-junction of Mg–7Gd–5Y–*x*Zn–0.5Zr alloys: (**a**) *x* = 0 wt%; (**b**) *x* = 3 wt%; (**c**) *x* = 5 wt%; (**d**) *x* = 7 wt%.

For the alloy III (Figure 11c) with LPSO phase as the main component and only a small amount of W-phase, many bridging traces can be observed on its cracks. For the alloy IV (Figure 11d) with the highest Zn content and only W-phase as the second phase, traces of grain boundary once well fed can be observed at the same position.

### 3.3.2. Effect of α-Mg Crystallization Precipitation on HTS

$T^{\alpha}_{coh}$ is the critical temperature for the solidification liquid feeding mechanism to change from global free feeding to local intergranular feeding, so the lower $T^{\alpha}_{coh}$ is, the lower HTS is. In addition, when the temperature is lower than $T^{\alpha}_{coh}$, the grain size will determine the size and thickness of the residual liquid film, as well as the solidification shrinkage stress of α-Mg, thus determining the HTS level of the alloy. The change of grain size and its variance with Zn content are shown in Figures 12 and 13. It can be seen that when Zn content was 0 wt%, 3 wt%, 5 wt% and 7 wt%, the average grain size was 23.5, 19.3, 17.6 and 14.9 μm, and the corresponding variance was 147.2, 76.9, 49.7 and 35.3, respectively.

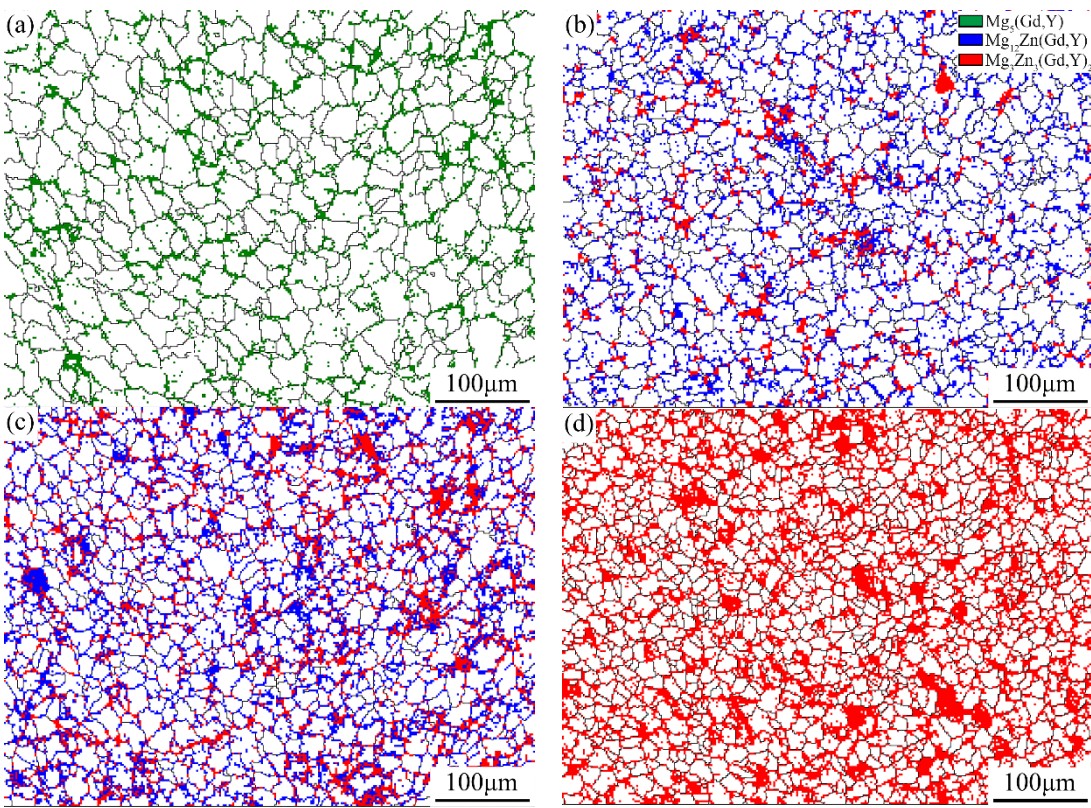

**Figure 12.** Grain morphology of Mg–7Gd–5Y–*x*Zn–0.5Zr alloy displayed by EBSD images: (**a**) *x* = 0 wt%; (**b**) *x* = 3 wt%; (**c**) *x* = 5 wt%; (**d**) *x* = 7 wt%.

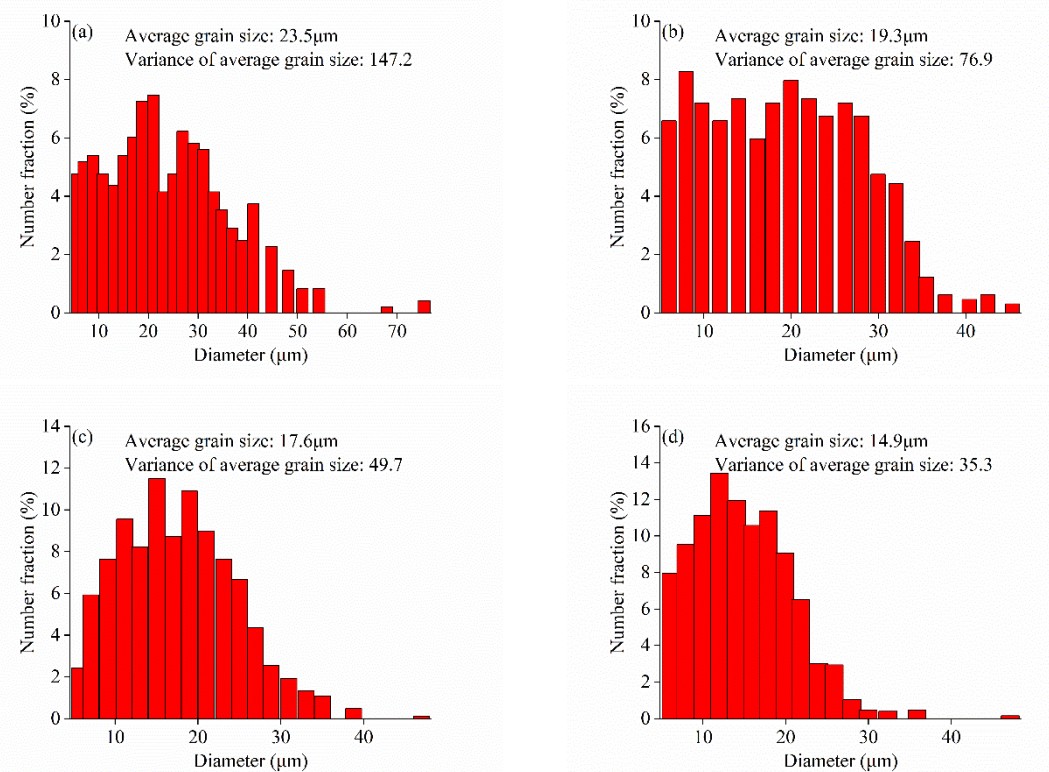

**Figure 13.** Distributions of grains size of Mg–7Gd–5Y–*x*Zn–0.5Zr alloys displayed by EBSD images: (**a**) *x* = 0 wt%; (**b**) *x* = 3 wt%; (**c**) *x* = 5 wt%; (**d**) *x* = 7 wt%.

As shown by Figure 12, green represents $Mg_5(Gd,Y)$ phase, blue represents LPSO phase and red represents W-phase. It was not difficult to see that the grain refinement of $\alpha$-Mg increased with the increase of Zn content. When $\alpha$-Mg crystallization has ended and the second phase has not yet precipitated, the residual liquid phase on the grain boundary also increased with the increase of Zn content. The mechanism of grain refinement with Zn content can be seen from the displacement of $\alpha$-Mg peaks in Figure 3 above. The calculated lattice parameters of $\alpha$-Mg vary with Zn content as shown in Table 4. It can be seen that the lattice constants a and c of $\alpha$-Mg decreased, while the axial ratio c/a increased, with the increase of Zn content, which indicated that the solid solubility of Gd and Y decreased with the increase of Zn content.

**Table 4.** Lattice constants of Mg–7Gd–5Y–*x*Zn–0.5Zr alloys.

| Alloy No. | Lattice Constants | | |
|---|---|---|---|
| | a (nm) | c (nm) | c/a |
| Alloy I | 0.32209 | 0.52116 | 1.61806 |
| Alloy II | 0.32162 | 0.52085 | 1.61946 |
| Alloy III | 0.32146 | 0.52081 | 1.62014 |
| Alloy IV | 0.32121 | 0.52078 | 1.62131 |

It can be seen from Figure 12 above that with the increase of Zn content, the total amount of the second phase precipitation increased, that is, the segregation of Gd and Y on the grain boundary should also increase. The atomic diameters of Gd and Y were larger than that of Mg, therefore this indirectly meant that the solid solubility of Gd and Y in $\alpha$-Mg decreased with the increase of Zn content, that is to say, more Gd and Y atoms need to diffuse to the grain boundary for $\alpha$-Mg nucleation and growth. This was bound to increase the nucleation rate of $\alpha$-Mg, reduce its growth rate and refine its grains. In addition, due to the asymmetry of the close packed hexagonal crystal structure, the distortion $\Delta$a caused by the solid solution of Gd or Y into the crystal lattice should be greater than $\Delta$c, so that the c/a axial ratio of $\alpha$-Mg increased with the increase of Zn content.

### 3.3.3. Effect of Second Phase Precipitation on HTS

It can be seen from Figure 5 above that $\alpha$-Mg was precipitated first, then $Mg_5RE$, LPSO or W second phase was precipitated, and the proportion of total precipitation of the second phase increased with the increase of Zn content. It can be seen that at the end of $\alpha$-Mg crystallization, the residual liquid phase on the grain boundary increased with the increase of Zn content, that is to say, the feeding capacity for the subsequent cooling shrinkage of $\alpha$-Mg also increased with the increase of Zn content, which made the HTS of the alloy decreased. With the decrease of temperature, when these residual liquid phases changed into solid phases, the latent heat released was directly proportional to the amount of transformation, and the heat released will alleviate the shrinkage stress of $\alpha$-Mg dendrite skeleton. Obviously, the HTS of the alloy will decrease with it.

Not only the amount of precipitation, but also the type of the second phase had a great influence on the HTS of the alloy. Different from Mg-Zn binary alloys containing low melting point second phase $MgZn_2$, when a certain amount of Zn was added to the alloy containing Gd and Y, high melting point LPSO or W phase will be formed, especially LPSO phase will precipitate before $\alpha$-Mg solidification shrinkage stress reaches the highest value, and it will bridge both sides of the grain boundary in the form of coherent lattice with the matrix to prevent the generation and expansion of hot tearing.

Figure 14 shows TEM observations and correspond EDS results and the selected area electron diffraction (SAED) patterns of the phases in Mg–7Gd–5Y–5Zn–0.5Zr alloy. Combined with the EDS results, the blocked-shaped phase had a composition with Zn/RE ratio of about 1, it was inferred that the block-shaped phase (marked by A in Figure 14a) consisted of 18 R LPSO phase. In addition, the fishbone-shaped phase (marked by B in Figure 14a) consisted of W-phase. The HRTEM images and SAED patterns shown in Figure 14c,d further demonstrated that the block-shaped phase was

18 R LPSO phase and the fishbone-shaped phase was W-phase with face-centered cubic structure. The accuracy of XRD results was also identified. It also can be seen from the SAED of LPSO phase, LPSO phase had a coherent lattice relationship with the $\alpha$-Mg matrix ($(0001)_{\alpha\text{-Mg}}//(0018)_{18R\text{-Mg}}$), which meant that LPSO phase at grain boundary played a strong pinning role on both sides of $\alpha$-Mg grain and inhibited hot tearing at the grain boundaries.

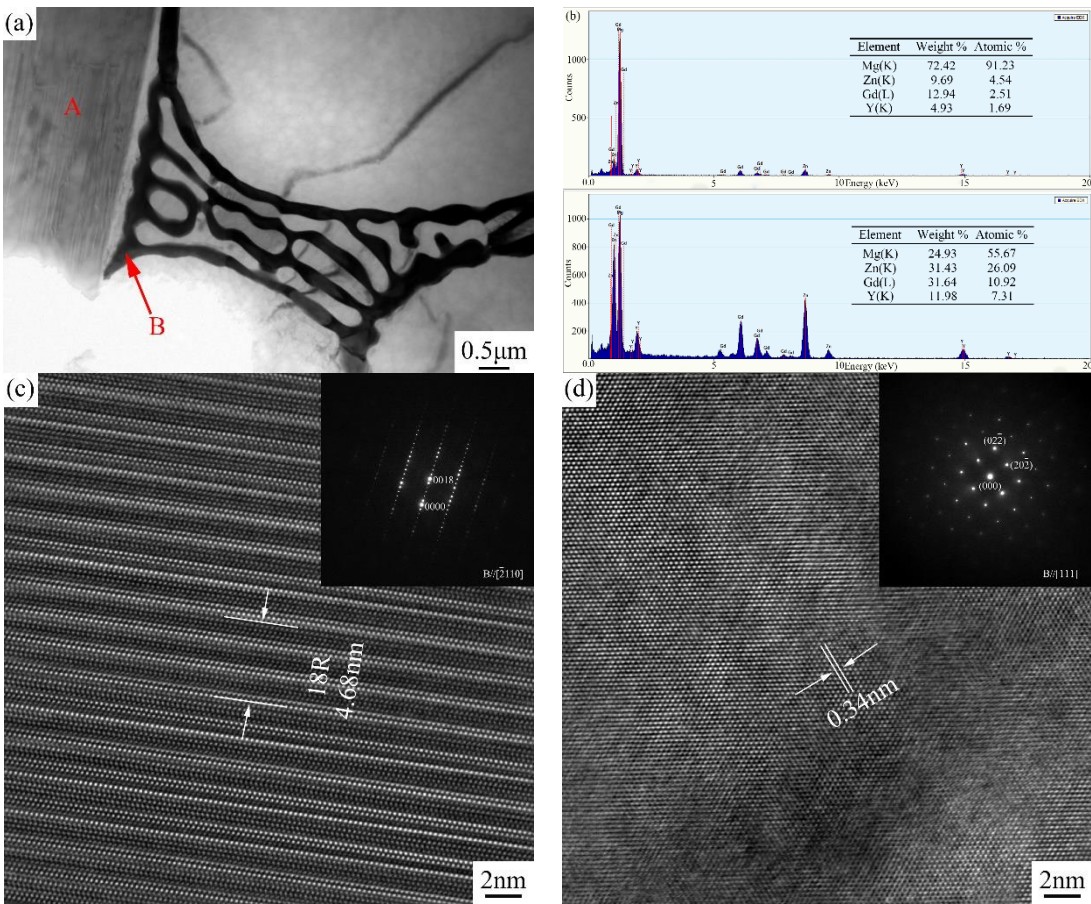

**Figure 14.** TEM observations and correspond EDS results, HRTEM images and selected area electron diffraction (SAED) patterns of the phases in Mg–7Gd–5Y–5Zn–0.5Zr alloy: (**a**) bright field (BF) image obtained from the alloy, (**b**) is the corresponding EDS results of the regions marked A and B, (**c**) and (**d**) are HRTEM images and corresponding SAED patterns of the phases.

## 4. Conclusions

Zn can significantly reduce HTS of Mg–7Gd–5Y–0.5Zr alloy, which almost linearly decreased with Zn content. When Zn content was 3 wt%, 5 wt% and 7 wt%, HTS will be reduced by 27%, 83% and 100%, respectively.

With the increase of Zn content, the solid solubility of Y and Gd in $\alpha$-Mg decreased, and the amount of solute interface segregation increased, which resulted in the change of crystallization mechanism of parent phase and the refinement of final grains. Macroscopically, it showed the increase of $T^{\alpha}_{coh}$, the shortening of inter-crystalline feeding stage, the decrease of total solidification shrinkage force and the decrease of HTS. The content of residual liquid phase and the type of precipitated phase will change with the content of Zn. Before the precipitation of the second phase, the increase of Zn content can increase the feeding ability of the residual liquid phase on the grain boundary. After the precipitation of the second phase, it can bridge the two sides of the grain boundary and improve the hot tearing resistance of the alloy, especially for the LPSO phase which is coherent with the grain boundary.

**Author Contributions:** Conceptualization, Z.W., S.L. and Z.L.; methodology, Z.W., S.L., Z.L. and F.W.; investigation, Z.W., X.W. and X.L. data curation, Z.W.; writing—original draft preparation, Z.W.; writing—review and editing, Z.W., S.L. and Z.L.; supervision, S.L., Z.L., F.W. and P.M. All authors have read and agreed to the published version of the manuscript.

**Funding:** This research was supported by the National Natural Science Foundation of China (No. 51571145), Liaoning Revitalization Talents Program (No. XLYC1807021) and Youth Project of Liaoning Education Department (No. LQGD2017032).

**Conflicts of Interest:** The authors declare no conflicts of interest.

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
