# Peer review of "Effects of Zn Content on Hot Tearing Susceptibility of Mg–7Gd–5Y–0.5Zr Alloy"

_metals, doi:10.3390/met10030414_

Round 1

Reviewer 1 Report

The work deals with quite interesting issues regarding the effects of Zn content on hot tearing susceptibility of 2 Mg–7Gd–5Y–0.5Zr alloy. However, the minor revision is still needed before the acceptance of this manuscript in Metals.

„The method of measuring Tα 88 coh has been described in detail elsewhere“. The method should be described in sufficient detail in the article. The reader should not look for additional literature.

Row 171, 218 and 219: Sentences „It can be seen that when Zn the content was 3 wt%, 5 wt%, and 7 wt%...“ and „When the content of Zn was 0, 3, 5 and 7 wt%...“ should be written in uniform.

Please explain the pic value in Figure 3.

It is written correctly? Figure 6 and Figure 9: „w(Z)/%” in the x-axis.

An analysis of research results needs to be expanded, for example 3.1.1.

Author Response

Response to Reviewer 1 Comments

Point 1: „The method of measuring Tα 88 coh has been described in detail elsewhere“. The method should be described in sufficient detail in the article. The reader should not look for additional literature.

Response 1: The detailed description of this method has been supplemented in the article (Row 97~100).

Point 2: Row 171, 218 and 219: Sentences „It can be seen that when Zn the content was 3 wt%, 5 wt%, and 7 wt%...“ and „When the content of Zn was 0, 3, 5 and 7 wt%...“ should be written in uniform.

Response 2: The writing of these two sentences and similar sentences has been unified.

Point 3: Please explain the pic value in Figure 3.

Response 3: Thank you very much for your questions. After repeated review, the author found that when interpreting the results of Fig. 3, there was indeed a representation beyond the experimental data. For this reason, the author carefully checked the diffraction peaks in Fig. 3, expressed the experimental results as objectively as possible, modified some words and marked them with the “Track Changes” function in Microsoft Word.

Point 4: It is written correctly? Figure 6 and Figure 9: „w(Z)/%” in the x-axis.

Response 4: The corresponding parts of the figures have been corrected. The modified figures are shown in the following (The specific pictures are displayed in the word document in the attachment).

Point 5: An analysis of research results needs to be expanded, for example 3.1.1.

Response 5: The analysis of research results has been expanded.

Reviewer 2 Report

The paper “Effects of Zn content on hot tearing susceptibility of 3 Mg–7Gd–5Y–0.5Zr alloy” is well written; it contains a great data value obtained with modern equipment.

Comments are present below.

1. Some references which are very interesting and important in this topic were missed, for example:

https://doi.org/10.1016/j.jma.2016.08.003

https://doi.org/10.1016/j.matdes.2009.12.003

https://doi.org/10.1016/j.msea.2010.07.086

About 32 refs from 38 are own or colleges from China Universities.

2. The effect of Zn content on HTS of magnesium was investigated before 1965 year by for example Professor Novikov I.I. and his group. The HTS drastically decreased with increasing Zn content from 2.5. to 7% and the alloys with a higher Zn content have not cracks. The first correlation of the Zn content effect is with an effective solidification range (ESR). ESR decreased and HTS decreased. The calculation of the ESR was suggested by Professor Zolotorevskiy V.S. group as applied in Al alloys. It is a very fast and simple method to determine the HTS by calculation of ESR using Sheil model without experimental work. It is very interesting to show the application of the suggested method in Mg alloys.

3.Lines 69-72. Are refs 22,23 illustrates a good description of the T-shaped method?

4. Line 91. Ref. 29 is not a Clyne-Davies paper.

5. 3.1.1 and 3.1.3 parts are better to analyze together.

Author Response

Response to Reviewer 2 Comments

Point 1: Some references which are very interesting and important in this topic were missed, for example:

https://doi.org/10.1016/j.jma.2016.08.003

https://doi.org/10.1016/j.matdes.2009.12.003

https://doi.org/10.1016/j.msea.2010.07.086

About 32 refs from 38 are own or colleges from China Universities.

Response 1: The above references have been added to the revised article and necessary discussion has been made.

Point 2: The effect of Zn content on HTS of magnesium was investigated before 1965 year by for example Professor Novikov I.I. and his group. The HTS drastically decreased with increasing Zn content from 2.5. to 7% and the alloys with a higher Zn content have not cracks. The first correlation of the Zn content effect is with an effective solidification range (ESR). ESR decreased and HTS decreased. The calculation of the ESR was suggested by Professor Zolotorevskiy V.S. group as applied in Al alloys. It is a very fast and simple method to determine the HTS by calculation of ESR using Sheil model without experimental work. It is very interesting to show the application of the suggested method in Mg alloys.

Response 2: Thank the reviewer for mentioning the literature about professional Novikov I.I. However, the author has not found this literature yet, and will continue to find it later. On the hot tearing research of Mg-Zn alloy, the author’s team has jointly published a paper [1] with Professor K.U. Kainer et al. from Helmholtz Association of German research centers. It is found that the curves of the hot tearing susceptibility of versus the content of Zn has a typical ‘Λ’ shape. With increasing content of Zn, the susceptibility of hot tearing first increases, reaches the maximum at 2~4 wt % Zn and then decreases again. In comparison, the alloy used in this study is multi-component alloy, so there are the following differences.

1. This study is also based on Clyne-Davies’ criterion, but the Newton baseline method based on thermal analysis test is used for liquid fraction instead of Scheil equation. This is because the basic data needed for Scheil equation applicable to multi-component alloy is not perfect, and Scheil model is based on some assumptions, such as the full diffusion of liquid phase at the solidification front, which will produce many errors, which is not as accurate as the Newton baseline method used in this paper [2].

2. For Mg-RE-Zn alloys, the mechanism of hot tearing is quite different from that of Mg-Zn binary alloys because of the complicated type of precipitated second phase. This paper is similar to Sweet’s study [3] on the effect of iron content on AA6060 alloy. It is found that the coordination of eutectic liquid feeding and early precipitation of Al-Fe-Si intermetallics has a great influence on the hot cracking sensitivity of the alloy. In other words, the ‘bridging’ mechanism of the second phase precipitation to the grain boundary cracking cannot be ignored.

3. It is a very fast and simple method to determine the HTS by calculation of ESR using Scheil model without experimental work for Binary alloy. However, there is another purpose in the study of thermal cracking, which is to understand the mechanism of its occurrence and try to control it. The author thinks that Clyne-Davies’ criterion is better than ESR criterion, because the former is a dimensionless quantity, which includes not only ESR part, but also the influence of parent phase crystallization path. For example, katgerman [4] developed a modified hot cracking standard as follows (The equation is displayed in the word document in the attachment).

Obviously, t0.99-tcr = α (TS-TCR); tcr-tcoh = β (TCR-Tcoh), where α and β are constants, TS ≈ t0.99 is the crystallization end temperature, TCR, that is, Tchi in this paper is the critical temperature in the brittle zone, and Tcoh is the dendrite coherent temperature. It can be seen that with the decrease of (TS-TCR), the decrease of temperature causes the decrease of the shrinkage in the brittle region, and with the increase of (TCR-Tcoh), the chance of liquid phase shrinkage increases in the channel blocked by the crystal. All of these results in the decrease of hot cracking sensitivity.

References

[1] Zhou, L.; Huang, Y.D.; Mao, P.L.; Kainer, K.U.; Liu, Z.; Hort, N. Influence of composition on hot tearing in binary Mg–Zn alloys. Int. J. Cast. Metals Res. 2011, 24, 170-176.

[2] Muthuraja, C.; Balasundar, I.; Ravi, K.R. Determination of Liquid Fraction in Mg–Zn–Y Alloys: Thermal Analysis Versus Thermodynamic Approach. Trans. Indian Inst. Met. 2018; 71, 2807-2811.

[3] Sweet, L.; Easton, M.A.; Taylor, J.A.; Grandfield, J.F.; Davidson, C.J.; Lu, L.M.; Couper, M.J.; Stjohn, D.H. Hot Tear Susceptibility of Al-Mg-Si-Fe Alloys with Varying Iron Contents. Metal. Trans. A 2013, 44A, 5396-5407.

[4] Katgerman, L.; A mathematical model for hot cracking of aluminum alloys during D.C. casting. JOM 1982, 34, 46–49.

Point 3: Lines 69-72. Are refs 22,23 illustrates a good description of the T-shaped method?

Response 3: Yes, these two references described the size, design significance and working principle of ‘T’-shaped mold in detail.

Point 4: Line 91. Ref. 29 is not a Clyne-Davies paper.

Response 4: The Ref. 29 has been corrected.

Point 5: 3.1.1 and 3.1.3 parts are better to analyze together.

Response 5: Parts 3.1.1 and 3.1.3 have been combined and analyzed together.

Reviewer 3 Report

This paper presents a good study on a Mg-Gd-Zn-Y alloy concerning the phases formation during cooling – providing an important basic for hot tearing. Only some minor issues should be addressed:

Abstract: there should not be abbreviation, HTS, LPSO

Introduction: please make sure, that you give background on the chemical composition of the LPSO phases

according to the alloy used: Mg12Zn(Gd,Y), same counts for the W-phase

Results: how many repeated measurements are done to evaluate the hot tearing parameters? For sure more than one time – please add error bars / standard deviation to parameters in Table 3

Figure 12: description of phases colored blue, red and green (Figure 12b top right) is very difficult to see

Figure 14b: labeling and scale too small

Author Response

Response to Reviewer 3 Comments

Point 1: Abstract: there should not be abbreviation, HTS, LPSO.

Response 1: The abbreviation in the abstract has been changed to the full name.

Point 2: Introduction: please make sure, that you give background on the chemical composition of the LPSO phases according to the alloy used: Mg12Zn(Gd,Y), same counts for the W-phase.

Response 2: The reviewer’s questions are very meaningful. It is not easy to find out the exact molecular formula of LPSO phase and W-phase compared with the alloy composition in this paper, nor the details that can be achieved in this paper. Therefore, the molecular formula of these two compounds in this paper is only in accordance with most of the current literature (newly added Ref. 13 in the paper), which does not affect the conclusions of this paper. By the way, the author has previously used HAADF-STEM to study the atomic structure of the LPSO phase in Mg–4.5Zn–6Y–0.5Zr alloy. In the dark field image of the high angle ring (see the figure in revised paper), the dark small dots are magnesium atoms, and the bright ones are the segregation of Y and Zn atoms on the stacking fault plane. However, no more detailed distinction has been made.

Point 3: Results: how many repeated measurements are done to evaluate the hot tearing parameters? For sure more than one time – please add error bars / standard deviation to parameters in Table 3.

Response 3: Thank you very much for the reviewer’s reminder. Each group of hot tearing parameters was obtained from at least three experiments. The test data can only be used in the article if the same experiment results appear three times. The corresponding description and standard deviation have been added to the article.

Point 4: Figure 12: description of phases colored blue, red and green (Figure 12b top right) is very difficult to see.

Response 4: The figure has been modified.

Point 5: Figure 14b: labeling and scale too small.

Response 5: The figure has been modified.
